# On the Close Relatedness of Two Rice-Parasitic Root-Knot Nematode Species and the Recent Expansion of *Meloidogyne graminicola* in Southeast Asia

**DOI:** 10.3390/genes10020175

**Published:** 2019-02-25

**Authors:** Guillaume Besnard, Ngan Thi-Phan, Hai Ho-Bich, Alexis Dereeper, Hieu Trang Nguyen, Patrick Quénéhervé, Jamel Aribi, Stéphane Bellafiore

**Affiliations:** 1CNRS-UPS-IRD, UMR5174, EDB, 118 route de Narbonne, Université Paul Sabatier, 31062 Toulouse, France; 2IRD, Cirad, University of Montpellier II, Interactions Plantes Microorganismes Environnement (IPME), 34394 Montpellier, France; ngan.phan-thi@ird.fr (N.T.-P.), alexis.dereeper@ird.fr (A.D.), patrick.queneherve@ird.fr (P.Q.), jamel.aribi@ird.fr (J.A.); 3IRD, LMI RICE, University of Science and Technology of Hanoi, Agricultural Genetics Institute, Hanoi, Vietnam; nthieu.biot@gmail.com; 4IOIT/USTH, VAST and UMMISCO, IRD, 18 Hoang Quoc Viet, Hanoi, Vietnam; hobichhai@gmail.com

**Keywords:** biological invasion, heterozygous genome, *Meloidogyne*, mitogenome, ribosomal DNA, root-knot nematode

## Abstract

*Meloidogyne graminicola* is a facultative meiotic parthenogenetic root-knot nematode (RKN) that seriously threatens agriculture worldwide. We have little understanding of its origin, genomic structure, and intraspecific diversity. Such information would offer better knowledge of how this nematode successfully damages rice in many different environments. Previous studies on nuclear ribosomal DNA (nrDNA) suggested a close phylogenetic relationship between *M. graminicola* and *Meloidogyne oryzae*, despite their different modes of reproduction and geographical distribution. In order to clarify the evolutionary history of these two species and explore their molecular intraspecific diversity, we sequenced the genome of 12 *M. graminicola* isolates, representing populations of worldwide origins, and two South American isolates of *M. oryzae*. *k*-mer analysis of their nuclear genome and the detection of divergent homologous genomic sequences indicate that both species show a high proportion of heterozygous sites (ca. 1–2%), which had never been previously reported in facultative meiotic parthenogenetic RKNs. These analyses also point to a distinct ploidy level in each species, compatible with a diploid *M. graminicola* and a triploid *M. oryzae*. Phylogenetic analyses of mitochondrial genomes and three nuclear genomic sequences confirm close relationships between these two species, with *M. graminicola* being a putative parent of *M. oryzae*. In addition, comparative mitogenomics of those 12 *M. graminicola* isolates with a Chinese published isolate reveal only 15 polymorphisms that are phylogenetically non-informative. Eight mitotypes are distinguished, the most common one being shared by distant populations from Asia and America. This low intraspecific diversity, coupled with a lack of phylogeographic signal, suggests a recent worldwide expansion of *M. graminicola*.

## 1. Introduction

Nematodes constitute an ancient and diverse animal phylum [1] representing 80% of multicellular animals on Earth [2]. Ubiquitous and found in various environments ranging from marine sediments to arid deserts, these metazoans can be carnivores, omnivores, bacterivores, or fungivores, or can feed exclusively on plants (15% of the described species). Plant parasitic nematodes (PPNs) cause an annual economic loss of over $US80 billion in agriculture worldwide [3]. Among PPNs, root-knot nematodes (RKNs, genus *Meloidogyne*) and cyst nematodes are the most important crop-damaging species. In Asia, RKNs are responsible for about 15% of the total economic losses in rice production [4].

Beside of sparse knowledge on nematode biodiversity, comprehension of the evolutionary history of this phylum is limited due to the rarity of plant parasitic nematode fossils discovered to date [5]. Interestingly, thanks to the few RKNs currently characterized, different modes of reproduction have been described, from amphimixis to mitotic parthenogenesis, with intermediate states where both sexual and asexual (parthenogenetic) reproduction coexist [6,7,8,9]. For instance, obligatory cross-fertilization was reported in *M. kikuyensis*, facultative meiosis (automixis) in *M. graminicola*, obligatory mitotic parthenogenesis (apomixis) in *M. oryzae*, and automictic or apomictic parthenogenesis in different polyploid or aneuploid forms, such as *M. arenaria*, *M. incognita* and *M. javanica* [9]. These variations in reproductive modes within RKNs were also related to specific ploidy levels and host range [7,9]. Based on a pioneering work on cytogenetic studies, Triantaphyllou [9] proposed that RKNs have undergone unparalleled extensive cytogenetic diversification. He also concluded that those features were characteristic of the establishment of meiotic and mitotic parthenogenesis in association with various degrees of polyploidy and aneuploidy.

During the last 20 years, DNA analysis of PPN genomic sequences has been a great help to taxonomists and evolutionists for analyzing and understanding the huge diversity and life history of PPNs and particularly RKNs. Indeed, comparative genomics has revealed huge genome diversity in RKNs resulting from hybridization, whole genome duplication, recombination, and horizontal gene transfers [10,11,12].

In RKNs, three main lineages (Clades I, II, and III) have been identified using the 18S ribosomal RNA gene as a marker [13]. Clade III includes meiotic parthenogenetic species (e.g., *M. exigua*, *M. graminicola*, *M. chitwoodi*) as well as the apomictic *M. oryzae*. Mainly present in Asia and tropical America, *M. graminicola* has become a worldwide major threat to rice agriculture, with recent records in Madagascar and South Europe [14,15]. In contrast, *M. oryzae*, the only described apomictic species belonging to Clade III also feeding on rice, is so far restricted to northern South America [16,17]. These two rice parasites have been discovered relatively recently, ca. 40–50 years ago [16,18] and they are known to display variable ploidy levels (i.e., 36 chromosomes in *M. graminicola* [6] and 54 in *M. oryzae* [19]). Because of their different levels of ploidy, reproductive patterns, and geographic distribution, the comparative study of *M. graminicola* and *M. oryzae* genomes in this narrow Clade III may provide important insights to understand *Meloidogyne*’s evolutionary issues.

Comparative genomics can shed light on the recent history of RKN species: trace back their spread, identify hybridization and genome duplication events, help to understand better the impact of their mode of reproduction on their demography, and help to investigate their adaptive processes to different environmental conditions [10,11,12]. Compared to other *Meloidogyne* species belonging to Clades I and II, the diversity and genetic structure of Clade III remains poorly understood, except for the facultative meiotic parthenogenetic *M. chitwoodi*, a well-known pest of potatoes [20,21]. Generating genomic data on both distinct species and isolates from various locations is thus necessary to better understand the origin and spread of this clade.

In the present study, we explore the genome structure and diversity of two related RKNs belonging to Clade III: *M. graminicola* and *M. oryzae*. Shotgun sequencing data were generated by HiSeq on 14 isolates. The genome complexity was investigated in both species with a *k*-mer approach, and the complete mitogenome and selected nuclear genomic DNA regions (including the nrDNA cluster and two low-copy genomic regions) were used to investigate the phylogenetic position of the two species within the *Meloidogyne* genus and explore the intraspecific diversity among *M. graminicola* isolates. Our results indicate a high proportion of heterozygous sites (ca. 1–2%) in the genome of the two investigated nematode species. In addition, as both species share highly similar nuclear sequences, we propose that *M. graminicola* is a potential parent of *M. oryzae*. Finally, the low genomic diversity observed within the studied *M. graminicola* populations strongly suggests a recent expansion of this taxon in Southeast Asia.

## 2. Materials and Methods

### 2.1. Nematode Sampling, DNA Extraction, and Sequencing

Twelve isolates of *M. graminicola* (*Mg*) and two isolates of *M. oryzae* (*Mo*) (Appendix A) were obtained after single nematode infection, as described in Bellafiore et al. [22]. Subsequently, each isolate was propagated in a hydroponic solution on the susceptible rice cultivar IR64 for four weeks before proceeding to egg and DNA extractions [22]. Briefly, juveniles and eggs were extracted from roots with a blender in a 0.8% hypochlorite solution, purified by centrifugation in a 60:40:20:10% sucrose discontinuous gradient to separate nematodes from different plant debris and limit contamination by other organisms such as bacteria [23], and finally rinsed several times with sterile distilled water and conserved at –80 °C. Total genomic DNA (gDNA) was isolated using proteinase K treatment and phenol-chloroform extraction followed by ethanol precipitation, as described by Besnard et al. [24]. DNA quality and quantification were assessed with a NanoDrop spectrophotometer (Thermofisher, Waltham, MA, USA) and PicoGreen^®^dsDNA quantitation assay (Thermofisher). One microliter of each gDNA extract was also loaded on a 1% agarose gel and subjected to electrophoresis to control the DNA integrity.

For each accession, 300 ng of double-stranded DNA (dsDNA) was used for shotgun sequencing using Illumina technology (San Diego, CA, USA) at the GeT-PlaGe core facility, hosted by INRA (Toulouse, France). DNAs were sonicated to get inserts of approximately 380 bp. The libraries were constructed using the Illumina TruSeq DNA Sample Prep v.2 kit, following the instructions of the supplier as previously described by Besnard et al. [24]. Libraries were multiplexed with libraries generated in other projects (24 per flow cell), and inserts were then sequenced from both ends on HiSeq 2000 or 2500 (Illumina).

### 2.2. Assembly and Phylogenetic Analysis of the Mitogenome

For each accession, the mitochondrial genome (mitogenome) was de novo assembled and annotated using the experimental procedure described by Besnard et al. [24]. As heteroplasmy was observed in a few cases, a consensus sequence was reconstructed for each nematode isolate, only considering sites as ambiguous (using IUPAC codes) when the minor sequence was supported by at least 5% of reads. Reads were then mapped onto the final sequence to assess the sequencing depth of the mitogenomes.

Reference mitogenome sequences for *M. graminicola* (NC_024275.1), *M. chitwoodi* (KJ476150.1), *M. enterolobii* (KP202351.1), *M. arenaria* (KP202350.1), *M. javanica* (KP202352.1), *M. incognita* (KJ476151.1), and *Pratylenchus vulnus* (GQ332425.1) were retrieved from GenBank (https://www.ncbi.nlm.nih.gov/genbank/). Six protein-coding genes (*cox1-cox3-nad4L-nad3-nad4- nad5*) and one ribosomal RNA (rRNA) gene (*rrnS*) were extracted from these mitogenomes. In addition, we retrieved these mitochondrial genes from *M. hapla* and *M. floridensis*, using a BLASTN search (http://xyala.cap.ed.ac.uk/downloads/959nematodegenomes/blast/blast.php; Appendix A). For each gene, sequences were aligned with those generated from the *M. graminicola* and *M. oryzae* isolates. All alignments were done using MUSCLE [25], and poorly aligned regions with the outgroup sequence (*P. vulnus*) were manually refined. The phylogenetic analysis of the sequence dataset was conducted and visualized based on maximum likelihood (ML) using MEGA v.7 [26] and confirmed using raxmlGUI V1.5 [27]. Initial trees for the heuristic search were obtained automatically by applying Neighbor-Join and BioNJ algorithms to a matrix of pairwise distances estimated using the ML approach, and then selecting the topology with the highest log likelihood value. The program JModelTest v.2.1.10 [28,29] was used to select the best-fit nucleotide substitution model by likelihood ratio test using Akaike Information Criterion and Bayesian Information Criterion. A general time reversible with discrete Gamma distribution (GTR + G) was used to model evolutionary rate differences among sites. A bootstrap analysis with 1000 replicates was performed to estimate the support for each node in the ML tree.

To infer the evolutionary history of *M. graminicola* on a maternally inherited genome, a haplotype network was then reconstructed based on the 13 mitogenome sequences available for this species, i.e., one from Brazil and 12 from Southeast Asia (including one Chinese isolate; sequence published and retrieved from GenBank [30]). An alignment was performed between all isolates excluding the 111R region (i.e., long region of the control region with 111-bp repetitions). Sequence polymorphisms were used to reconstruct a reduced-median haplotype network with Network v.5 [31]. The presence or absence of indels was coded as 1 and 0, respectively. A short inversion was similarly coded. Mutations leading to heteroplasmic sites were further considered and also positioned on the network.

### 2.3. Assembly and Phylogenetic Analysis of Nuclear Ribosomal DNA Sequences

The tandemly repeated nature of the nrDNA cluster associated to concerted evolution between adjacent copies is usually an advantage to properly assemble this nuclear genomic region with low-depth sequencing data [32]. Complete nrDNA sequences (including 18S-ITS1-5.8S-ITS2-28S) were thus reconstructed using the approach described in Besnard et al. [24]. Briefly, a sequence containing Internal Transcribed Spacers (ITS) from *M. graminicola* or *M. oryzae* (GenBank Nos. KF250488 and KY962653) was used as a seed and elongated by recursively incorporating reads identical to at least 90 bp at the ends of the contig, using Geneious v.6 (https://www.geneious.com). Diverging nrDNA units were however found in each isolate, and needed to be assembled separately for proper phylogenetic analyses. Paired-end reads were thus carefully phased after visual inspection, resulting in the assembly of two homologous sequences in each accession.

Phylogenetic analyses were then conducted to investigate relationships between *Meloidogyne* species and decipher the origin of diverging units in *M. graminicola* and *M. oryzae* genomes. We selected nrDNA sequences from GenBank for seven *Meloidogyne* species that belong to different *Meloidogyne* lineages of Clade III (i.e., *M. trifoliophila*, *M. chitwoodii*, *M. naasi*, *M. fallax*, *M. minor*, *M. graminis* and *M. marylandii*). Two trees were reconstructed using the D2–D3 region (656 bp) from the 28S gene, and the ITS (538 bp). Whole nrDNA sequences (ca. 8 Kb) of *M. graminicola* and *M. oryzae* were further aligned to investigate phylogenetic relationships between their divergent units. All alignments were done using MUSCLE [25] and cleaned with Gblocks using default settings [33]. Phylogenetic analyses were done using MEGA v.7 [26] and raxmlGUI V1.5 [27]. Phylogenetic trees were reconstructed using the ML method based on the GTR + G model selected by JModelTest v.2.1.10 [28,29]. Support of nodes was estimated with the rapid bootstrap algorithm (1000 iterations).

### 2.4. Nuclear Genome Structure Analysis

A draft genome and a transcriptome of *M. graminicola* were recently published [34,35]. However, the draft genome of this species is incomplete (84.27% CEGMA, (Core Eukaryotic Genes Mapping Approach) [36]) without any description on genome structure and organization [34]. We further tested the completeness of published draft genome and transcriptome of *M. graminicola* using BUSCO v.3 (Benchmarking Universal Single-Copy Orthologs) [37]. The nematode species *Caenorhabditis elegans* was set as the species-specific-trained parameters for gene prediction. The BUSCO dataset “Eukaryota *odb9*”, which includes 303 Eukaryote single-copy orthologs, was used as the reference.

The sequencing reads of the 14 isolates were cleaned and contaminant sequences (e.g., from bacteria) were removed. FastQC [38] was used for quality control. Preprocessing procedure included adaptor removal, quality trimming, duplication and contaminant removal. In details, Trimmomatic [39] was employed to trim reads of TruSeq adaptors and low-quality nucleotides (cut-off score of 20). Duplication levels were reduced to less than 15% by FastUniq [40]. On close inspection of GC content plots, minor second peaks of 50–70% were found, indicating potential contaminants. We hence utilized BBTools (Joint Genome Institute, University of California, CA, USA; https://jgi.doe.gov/data-and-tools/bbtools/) to filter and remove reads that mapped over 80% identity to reference genomes (Appendix A). Detailed number of reads for each isolate before and after contamination removal are listed in Appendix A. After contamination removal, eight *M. graminicola* (i.e., Mg-VN6, Mg-VN11, Mg-VN18, Mg-VN27, Mg-L1, Mg-C21, Mg-P, Mg-Brazil) and the two *M. oryzae* isolates showed only one peak of GC content in genome sequences. Thus, clean reads from these isolates were then used for further analyses. Four *M. graminicola* isolates (i.e., Mg-L2, Mg-C25, Mg-Java, Mg-Borneo) were removed due to the presence of a second peak in GC content plots.

The overall characteristics of *M. graminicola* and *M. oryzae* nuclear genomes were determined by a *k*-mer analysis. We investigated the Mg-VN18 and Mo-M2 isolates, which have similar sequencing coverage (see below and Appendix A of the Appendix A). From the cleaned reads, we extracted canonical *k*-mers (*k* from 15 to 31 with step 2) by Jellyfish [41] to plot *k*-mer distributions of the two isolates. For each *k*, the abundance histograms and corresponding genome estimates were investigated (Appendix A shows four different *k* values on genome sequences of eight *M. graminicola* isolates and two *M. oryzae* isolates). Since similar shapes and estimates were found across *k* values, we opted to discuss here the 25-mer analyses of genome structure and complexity. *k* of 25 was also chosen to characterize the genomes of human [42] and plants (e.g., maize [43]), owing to its robustness to repeated region and to sequencing error. More concretely, coverage thresholds of heterozygous and homozygous peaks and repeated regions in the distributions were manually detected, according to Kajitani et al. [44]. Based on those, sequencing coverage, genome size, heterozygous rate and repeated proportions were then calculated as described in the Appendix A.

### 2.5. Analysis of Two Low-Copy Nuclear Genomic Regions Among Accessions of *Meloidogyne graminicola* and *Meloidogyne oryzae*

In order to determine the origin of the high heterozygosity observed in *M. graminicola* and *M. oryzae* (see below), we decided to investigate the diversity of low-copy regions from the nuclear genome. A reference-guided approach was used to de novo assemble (in Geneious) two genomic regions containing microsatellite motifs (further referred to TAA6 and ACC6 regions). For each region, two distinct homologous sequences were clearly present in the 12 *M. graminicola* accessions, while two or three variants were detected in *M. oryzae*. Contigs of about 6 Kb were first assembled in Mg-VN18 and Mo-M2 by elongating initial sequence, recursively incorporating reads identical to at least 30 bp at both ends of the contig. Paired-end reads were carefully phased after visual inspection, resulting in the assembly of two or three homologous sequences. Their homology was estimated with MEGA (1 – *p*-distance). For each accession, reads of *M. graminicola* or *M. oryzae* were finally mapped onto the assembled contigs, majority-rule consensus sequences were extracted and sequencing depth was estimated. These final assemblies of each homologous sequence were manually checked to identify potential heterozygous sites, i.e., single nucleotide polymorphisms (SNPs) supported by at least two distinct reads. To identify coding sequences in these nuclear contigs, BLAST (https://blast.ncbi.nlm.nih.gov/) alignments were finally performed, using a database of 66,396 *M. graminicola* transcripts [35], considering each genomic region as a query.

All distinct homologous sequences of these two genomic regions were finally used to investigate the phylogenetic relationship between *M. graminicola* and *M. oryzae*. Sequence alignment and phylogenetic analysis methods were done as described in Section 2.3. The GTR + G and GTR + I models were applied for ML analyses of TAA6 and ACC6 genomic regions, respectively.

## 3. Results

### 3.1. General Mitogenome Features of *Meloidogyne oryzae*

We generated 8,559,374 and 8,316,341 paired-end 100-bp reads on the two *M. oryzae* isolates from French Guiana (M1) and Suriname (M2), respectively (Appendix A). Based on these data, two mitogenomes of *M. oryzae* were assembled, the first produced for this species (GenBank No: MK507908 and LR215847). These sequences are AT-rich (83.2%) as previously reported in other *Meloidogyne* spp. [21,24,45]. Like in *M. graminicola*, tandemly repeated elements of 111 bp (111R) are found in the control region [24]. In contrast, the other region with repeated elements (namely the 94R region) in the *M. graminicola* mitogenome shows shorter repeated elements in *M. oryzae* (65R). The 94R region corresponds to a 65R region with the addition of 29 bp at the end of the 65-bp elements (Appendix A). Note that the repeated 65-bp element is different from the 63-bp element of the 63R region present in the mitogenomes of the MIG group [*M. incognita* group ([46]]; Appendix A).

Excluding the 111R region, the mitogenome length of the two *M. oryzae* isolates is estimated at 17,069 bp (Mo-M1) and 17,066 bp (Mo-M2). Thirty-six genes were identified in these *M. oryzae* mitogenomes (12 protein-encoding genes, 22 transfer RNA genes and two rRNA genes), with the exact same gene arrangement observed in *M. graminicola* (Appendix A). Like in *M. graminicola*, one large non-coding region (NCR) was also found in *M. oryzae* between *nad4* and the tRNA gene *S_2_*.

Alignment of the two *M. oryzae* mitogenome sequences without the 111R region revealed 11 nucleotide polymorphisms, including six substitutions (SNPs) and five indels (insertion/deletion) (Appendix A). Eight of these polymorphisms are located in non-coding regions, while the other three are SNPs located in *apt6*, *nad5*, and *cox3*. The two SNPs detected in *apt6* and *nad5* are non-synonymous and the SNP present in *cox3* is a silent mutation. Alignment of *M. oryzae* (Mo-M2) and *M. graminicola* (Mg-P) mitogenomes (excluding the 111R region) revealed a total of 619 SNPs and 58 indels, for an average identity of 96.4% (Appendix A).

### 3.2. Phylogenetic Analyses of *Meloidogyne* Based on Available Complete Mitogenomes

The phylogeny of the *Meloidogyne* genus reconstructed from seven mitochondrial genes supports the distinction of the three main RKN lineages (Figure 1). The early diverging lineage corresponds to Clade I, with *M. enterolobii* sister to a clade of four closely related species (*M. arenaria*, *M. javanica*, *M. incognita* and *M. floridensis*). Clade II, which is only represented by *M. hapla*, is placed next to Clade I, but with a low support (bootstrap value of 78). In Clade III, *M. chitwoodii* is sister to *M. oryzae* and *M. graminicola*, with all nodes strongly supported.

### 3.3. *Meloidogyne graminicola* Haplotype Diversity

The mitochondrial DNA variability was investigated on 13 *M. graminicola* isolates. Eleven new mitogenome sequences were assembled in our study (GenBank Nos: LR215848-LR215858) and were aligned with the two sequences already available in GenBank [NC_024275 (Philippines) and KJ139963 (China)]. These sequences, excluding the 111R region, range from 16,808 bp (Mg-Borneo) to 16,905 bp (Mg-C25 and Mg-VN27; Appendix A). Fifteen polymorphisms were detected, including 11 SNPs, three indels and one inversion (Appendix A; Figure 2). Eight of these 15 polymorphisms were located in coding regions and seven in the NCR (Appendix A). Three substitutions were transversions (A, G ↔ C, T) and eight were transition mutations (A ↔ G, C ↔ T) (Appendix A). One indel in *atp6* and four SNPs in *nad5*, *cox1*, *nad1*, and *cox3* were non-silent mutations. The 15 polymorphic sites allowed for the distinction of eight haplotypes among the 13 isolates (Figure 2). Eight heteroplasmic sites were also detected in five isolates (Mg-L1, Mg-Borneo, Mg-VN6, Mg-VN11, and Mg-VN18; Appendix A; Appendix A). Four of them were non-synonymous and located in *cox1* and *cob*. Taken together, these intraspecific polymorphisms allowed distinction of 12 haplotypes among the 13 isolates analyzed. Although polymorphisms were phylogenetically non-informative (except an insertion of one 94-bp repeat in the 94R region in Mg-C25 and Mg-VN27, that is putatively homoplasic; Appendix A), we can assume that the likely ancestral sequence represents the most frequent haplotype in the center of the network. Only Mg-Brazil and Mg-Java share this haplotype without any heteroplasmic sites (Appendix A; Appendix A).

### 3.4. Analysis of Nuclear Ribosomal Genes

Complete nrDNA sequences (including NC1-5S-NC2-18S-ITS1-5.8S-ITS2-28S; ca. 8 Kb) were reconstructed for each isolate (GenBank Nos.: LS974432, LS974433, LS974439, and LS974440). Two divergent nrDNA sequences (or ribotypes) were always detected (with a mean sequence identity of ca. 99%) in all *M. graminicola* isolates. Two main types were also assembled from the *M. oryzae* isolates (with a mean sequence identity of 99%), but a few intra-individual polymorphisms were detected in one of them. In both species, sequence polymorphisms (SNPs and Indels) between two nrDNA types were mostly located in non-coding regions (NC1, NC2 and ITS; Appendix A). No polymorphism was detected between individuals on each nrDNA type among all *M. graminicola* isolates or among the two *M. oryzae* isolates. For phylogenetic reconstructions, we consequently considered only one pair of nrDNA sequences to represent *M. graminicola* and *M. oryzae*. Since complete nrDNA sequences have not been released for other *Meloidogyne* species belonging to Clade III, only the D2–D3 (656 bp) and ITS (538 bp) regions were independently used to analyze phylogenetic relationships within this root-knot nematode lineage. The two phylogenetic trees reconstructed on these regions were similar (Figure 3A and Appendix A) and sustain relationships that are currently recognized in this *Meloidogyne* clade [47,48,49]. Three subclades were found where *M. marylandi* and *M. graminis* form an early diverging group. The *M. chitwoodi* group (*M. minor*, *M. chitwoodi*, and *M. fallax*) forms a second cluster, sister to the *M. graminicola* group. In this latter, *M. naasi* forms an early-diverging lineage, while *M. trifoliophila* and the two nrDNA types of *M. graminicola* and *M oryzae* were intermingled in a well-supported clade (92% bootstrap). Interestingly, as strongly supported by the phylogenetic analysis from the whole nrDNA sequence, ribotypes I of *M. graminicola* and *M. oryzae* were almost identical, in contrast to their ribotype II that appeared relatively different from each other (Figure 3A–B and Appendix A).

### 3.5. Nuclear Genome Features of *Meloidogyne graminicola* and *Meloidogyne oryzae*

The genome assembly completeness measured with BUSCO (Appendix A) showed that only 73.6% of a set of conserved Eukaryote genes are fully present on the published draft genome [34]. More precisely, 15.2% of genes are partially mapped, while 11.2% are completely missing. In comparison, the transcriptome assembly completeness [35] is higher with the detection of 88.1% of Eukaryote genes (Appendix A) but still incomplete.

The Mg-VN18 (*M. graminicola*) and the Mo-M2 (*M. oryzae*) samples contain 8.4 million and 7.6 million of paired-end reads, respectively. The two genomes have similar GC contents, 24% in *M. graminicola* and 26% in *M. oryzae*. The equivalent sequencing coverage for both samples (ca. 21x; see Appendix A in the Appendix A) allows genome comparison analyses. Under the assumption that genomic *k*-mer profile follows Poisson distribution [50], the peaks can be used to infer about genome ploidy. We noted a bimodal *k*-mer distribution for the *M. graminicola* genome (Figure 4 and Appendix A), a major peak at sequencing coverage of 16x and a minor peak at 32x, which correspond respectively to heterozygous and homozygous peaks. Similar bimodal *k*-mer pattern has been previously described in other species and the two peaks found for *M. graminicola* could theoretically indicate either an allotetraploid genome, or a diploid genome with a high proportion of heterozygous sites (>1%) [44,51]. In *M. oryzae,* the *k*-mer distribution is more complex with a heterozygous peak at 16x and a heavy tail up to 60x, as expected with an increased ploidy level [19]. In addition, a higher proportion of repeats was found in the *M. oryzae* genome (6.85% versus 5.92% for *M. graminicola*). Based on the hypothesis of a diploid *M. graminicola*, and a triploid *M. oryzae* [6,19], the haploid genome sizes could be estimated at about 35.64 Mb and 37.57 Mb respectively. These sizes are slightly lower than the haploid genome sizes of other *Meloidogyne* spp. (40–50 Mb) but in agreement with previous studies suggesting a smaller genome for *M. graminicola* [6,52].

### 3.6. Analysis of Two Low-Copy Nuclear Regions in *Meloidogyne graminicola* and *Meloidogyne oryzae*

Two nuclear low-copy genomic regions of approximately 6.3 Kb were assembled (named ACC6 and TAA6; Genbank Nos: LS974437, LS974438, LS974442, LS974443, and LR131833 to LR131839). For both genomic regions, two divergent copies (arbitrarily named Type I and Type II) were detected with an even sequencing depth in all accessions of *M. graminicola*. The sequence identity between the two types was 97.61% and 98.36% for ACC6 and TAA6, respectively (Appendix A). Among isolates, the mean sequencing depth of these genomic regions was between 9.4x and 22.2x if we exclude Mg-Java (0.9x; Appendix A). On average, the sequencing depth of a single-copy region of the nuclear genome was thus 93 (^+^/_−_ 40) times lower than for the mitogenome (Appendix A). No sequence polymorphism and no recombination between Types I and II were detected on a total of ca. 25.2 Kb among 11 isolates of *M. graminicola* (Mg-Java was excluded from this analysis because sequencing depth was too low and some parts were missing). Two open reading frames, encoding GPI ethanolamine phosphate transferase 1 and an unknown protein, were annotated in TAA6, while no gene was detected in ACC6.

Homologous sequences of the ACC6 and TAA6 genomic regions were also found in *M. oryzae* (and their different sequence variants further named Types I, III and IV; see below). Sequence identity between two types of sequences was around 98% (97.81 to 98.14% and 98.07% for ACC6 and TAA6, respectively (Appendix A)). For TAA6, two homologous sequences were detected with one showing a sequencing depth approximately twice of the other (Appendix A). Note that the most abundant type was not the same in Mo-M1 (Type I) as in Mo-M2 (Type III). In the case of ACC6, two copies were detected in Mo-M1, with Type III twice more abundant than Type I (Appendix A), whereas in Mo-M2, three distinct ACC6 copies were assembled, each type with a similar sequencing depth (10.1-11x; Appendix A). At each nuclear genomic region, the presence of three copies present in equal abundance or of two copies with one twice as abundant as the other is the expected pattern for a triploid genome. Importantly, in contrast to *M. graminicola* accessions, the genetic profile of the two *M. oryzae* accessions is different (Appendix A), indicating genome reshuffling in this species.

For each genomic region, the phylogenetic analysis of the different copies isolated from *M. graminicola* and *M. oryzae* showed that Type I of both species is almost identical (Figure 3C–D; Appendix A). On the other hand, Type II is specific to *M. graminicola*, while Types III and IV are only detected in *M. oryzae* isolates (Figure 3D). At the interspecific level, the different types of sequences show an identity comprised between 97.62 and 98.38% (Appendix A), so not higher than at the intra-individual level (see above).

## 4. Discussion

In the present study, we generated new genomic resources for two species of *Meloidogyne* (*M. graminicola* and *M. oryzae*). Genomes containing distinct nuclear ribosomal DNA units and low-copy regions were reported for the first time in *Meloidogyne* species belonging to Clade III. Our data also provided phylogenetic patterns revealing a close relationship between the two investigated species (with one possibly a parent of the other), as well as evidences for their highly heterozygous genomes and a recent worldwide expansion of *M. graminicola*.

### 4.1. *Meloidogyne oryzae* and *Meloidogyne graminicola* Are Closely Related Root-Knot Nematodes

Sequence analyses of both mitochondrial and nuclear genomes demonstrated that, despite different mode of reproduction, *M. graminicola* and *M. oryzae* are closely related species. In early reports based solely on partial mitochondrial genomic regions [12], *M. graminicola* and *M. oryzae* were already presented as sister species. This was confirmed by our different mitogenomic analyses, where a high identity (96.8%) as well as a highly conserved mitogenome structure were shown between both species (Appendix A). Moreover, analysis of the complete nrDNA cluster together with two low-copy nuclear regions supported not only the existence of two divergent types of sequences in *M. graminicola*, but also at least two or three types in *M. oryzae.* On each of these genomic regions, one type of sequences was shared by *M. graminicola* and *M. oryzae* (Figure 3; Appendix A) strongly suggesting a closely related ancestor between the two species. Furthermore, this pattern suggests that *M. graminicola* (36 chromosomes) has played the role of parent of *M. oryzae* (54 chromosomes).

### 4.2. High Heterozygosity of *Meloidogyne graminicola* and *Meloidogyne oryzae* Genomes

With the recent release of RKN nuclear genomes [12,53,54], comparative genomics studies have been done on a few species [12,55], and hybridizations involving whole genome duplications (i.e., allopolyploidization) were underlined in the mitotic parthenogenetic species such as *M. incognita*, *M. arenaria*, and *M. javanica* [53,56]. This was also suggested for the obligate meiotic parthenogenetic *M. floridensis* [11] but conversely, the facultative sexual *M. hapla* did not show genome duplication [36,38]. Therefore, allopolyploidization events seem to be associated to apomictic RKNs [11,12,57,58].

Thanks to the presence of two distinct types of homologous sequences (diverging by ca. 2% for low-copy regions) at each nuclear region investigated here, we show strong evidence for a highly heterozygous genome of *M. graminicola*. To date, this has never been reported in facultative meiotic parthenogenetic nematodes. First, two nrDNA types as well as two types of low-copy genomic regions (ACC6 and TAA6) were found in all *M. graminicola* isolates (see above; Figure 3). Secondly, as reported in other species [44,51,59], the *k*-mer distribution in the *M. graminicola* genome (Figure 4) suggests a high proportion of heterozygous sites (>1%). To explain this, homologous regions may harbor two divergent genomic copies that are expected to generate two *k*-mer peaks: a first peak (or hetero peak; here at 16x) and a second one with a two-folds coverage (or homo peak; here at 32x) that respectively result from non-conserved and conserved segments of the genomic region (see [60]). This is important to consider when automatically reconstructing the genome draft sequence of this species [34], because the assembly algorithm may encounter difficulties in regions with a high proportion of heterozygous sites, especially, as here, when using short-read sequences (i.e., HiSeq). In addition, some homologous regions may be separately assembled while others could be merged in a consensus sequence. Thus, this may explain the lower quality assembly of the recently published *M. graminicola* genome [34] than in other *Meloidogyne* spp. genomes (Appendix A; [56]). A sequencing strategy based on both long and short reads will be necessary to resolve this issue. Furthermore, we did not detect any polymorphic sites in each homologous sequence of three nuclear regions (nrDNA, TAA6, ACC6) among 11 *M. graminicola* isolates. We also did not detect any recombination events between homologous sequences of the TAA6 and ACC6 regions. The maintenance of the same genetic profile (Appendix A) among isolates collected in Asia and America suggests no reassortment of homologous chromosomes during the reproduction since the worldwide spread of *M. graminicola*, probably as a consequence of a preferential asexual reproduction in this species. In addition, our results do not confirm previous works that have reported sequence ITS polymorphisms among *M. graminicola* isolates [22,61]. The observation of two distinct nrDNA types in *M. graminicola* (Figure 3A–B) indicates that this genomic region has to be carefully used when reconstructing phylogenies of *Meloidogyne*. Indeed, the coexistence of divergent ribotypes within individuals and chimerical sequences generated by PCR may explain the high ITS variation previously reported [22,61].

Like *M. graminicola,* the apomictic *M. oryzae*, also belongs to Clade III in the nematode classification and shows evidence for a more complex genome (Figure 4). The flattened curve of *k*-mer distribution may reflect the presence of three peaks at ca. 16x, 32x, and 48x resulting from the presence of three distinct genomes, due to its triploid status [19]. These results confirm that the *M. oryzae* genome is more complex that the *M. graminicola* genome, due to an additional set of chromosomes (54 vs. 36 chromosomes [8,9]). In addition, two or three types of diverging sequences were detected in the low-copy regions ACC6 and TAA6 (Figure 3C–D; Appendix A). Sequencing depth of each sequence type (Appendix A) is compatible with the supposed triploid status of *M. oryzae* [8,9], but the distinct genetic profile of the two analyzed accessions is a strong evidence for genome reshuffling in *M. oryzae*. This latter result is surprising since the species is reported as reproducing by mitotic parthenogenesis [9], which should not allow homologous chromosome reassortment. We can thus hypothesize that, following the polyploidy event, at least a few meiosis events have been involved in the diversification of this species.

### 4.3. *Meloidogyne graminicola* Isolates Present Very Low Variation and an Absence of Phylogeographic Pattern, Suggesting a Recent Worldwide Expansion

Both nuclear and mitochondrial genomic sequences were analyzed among 12 isolates of *M. graminicola* that were collected from several geographical origins. The mitogenome is known to evolve more rapidly that the nuclear genome in the Nematoda phylum, so it can be used to study the population history of *Meloidogyne* spp. [20,62]. As mentioned above, no sequence polymorphism was detected within the nuclear genomic regions (25.2 Kb of low-copy regions plus 15.4 Kb of nrDNA regions), but low mitochondrial diversity was also revealed. To explain such a low degree of polymorphism between *M. graminicola* isolates, we propose two hypotheses: the first one is that this species accumulates mutations very slowly due to an uncommon DNA replication apparatus as was suggested for instance in coral [63] and turtle [64]. The second hypothesis is that *M. graminicola* recently spread at worldwide level, as supported by the mitochondrial haplotype network (Figure 2). Indeed, this analysis showed that the most common, ancestral haplotype (from which the other types are derived) is shared by distant populations from Brazil and Southeast Asia. Interestingly, *M. graminicola* is the only species of Clade III showing a widespread distribution, from America to Asia with recent reports in Madagascar [14] and South Europe [15]. Moreover, this species presents a singular adaptation to different rice field environments (see map distribution of *Meloidogyne* spp. on https://gd.eppo.int/). We could hypothesize that due to its virulence against rice (*Oryza sativa*), *M. graminicola* was able to rapidly spread over the world due to human activities that allowed long-distance translocations. Considering that rice domestication started between 13,500 and 8000 years ago, with three separate diversification centers in Asia [65,66], it is questionable to assert that the pathogen was originally present in today’s rice cropping areas. Indeed, if the worldwide distribution of *M. graminicola* was older than rice agricultural development, we could expect much larger variation in the mitochondrial genome and a phylogeographic structure indicating an ancient expansion of the species. Agricultural practices such as rice transplanting and plantation or banana suckers from infected nurseries seem like good ways to spread the pathogen [4,67]. However, the absence of *Meloidogyne* fossils and the lack of evaluation of evolution rates in this species prevent a clear estimation of this event. Resolving the recent expansion of *M. graminicola* populations will probably require an exhaustive nuclear genome analysis, but this will need first to generate a high-quality genome assembly with a clear distinction of homologous sequences that can be used as a reference for detecting SNPs.

## Figures and Tables

**Figure 1 genes-10-00175-f001:**
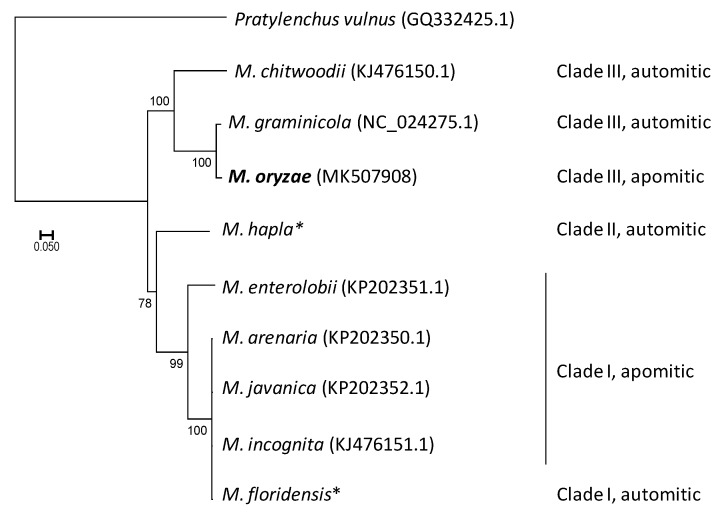
Maximum likelihood (ML) phylogenetic tree of *Meloidogyne* spp. based on seven mitochondrial genes (*cox1-rrnS-cox3-nad4L-nad3-nad4-nad5*) from nine root-knot nematode species. The tree was rooted with *Pratylenchus vulnus* as an outgroup. GenBank (https://www.ncbi.nlm.nih.gov/genbank/) mitogenome reference sequences are indicated in parenthesis (with sequence from this study in bold), except for *M. hapla* and *M. floridensis* (*) for which accession numbers corresponding to the different genes used in the phylogeny are given in Appendix A. The numbers beside branches represent ML bootstrap support values >50%. Scale bar represents substitutions per nucleotide position.

**Figure 2 genes-10-00175-f002:**
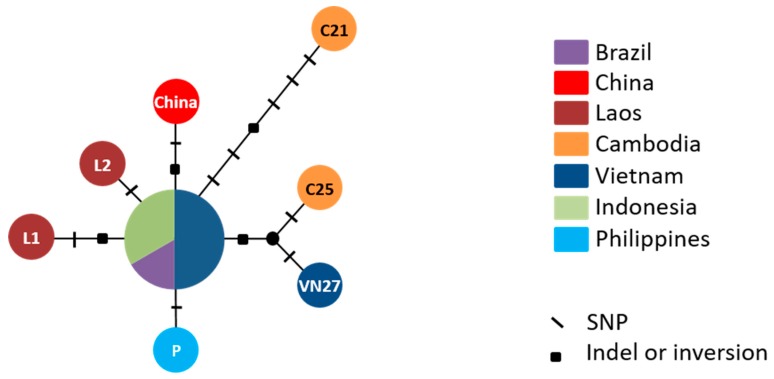
Reduced-median network of *Meloidogyne graminicola* mitochondrial haplotypes. The network was reconstructed with Network v.5 [30], using the 13 available mitogenome sequences, excluding the 111R region. Code names of distinct populations are indicated in the circles and geographic origin is displayed by different colors. The number of mutations is shown on the branches with slashes and black squares that respectively indicate SNPs, and indels or inversion sites. See Appendix A for the network considering heteroplasmic sites.

**Figure 3 genes-10-00175-f003:**
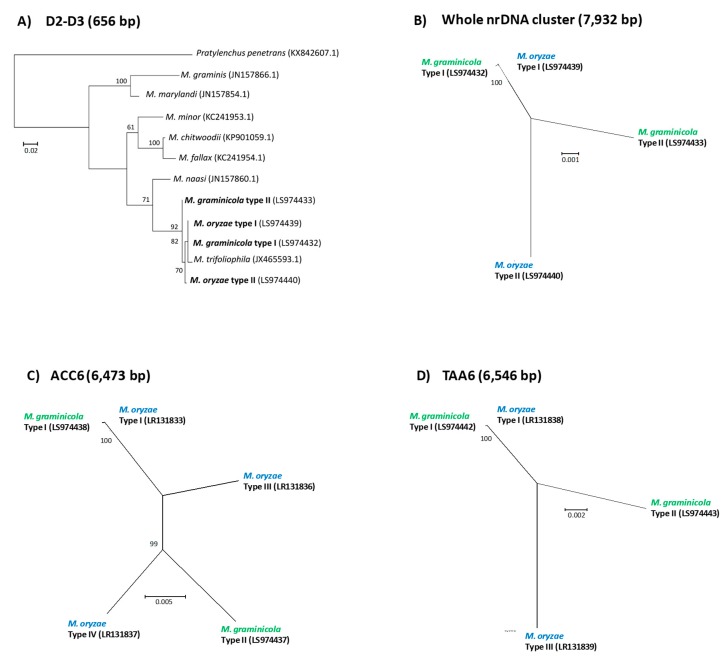
Maximum likelihood phylogenetic reconstructions of *Meloidogyne* species belonging to Clade III, based on (**A**) D2–D3 nrDNA sequence, (**B**) the whole nrDNA cluster, (**C**) the ACC6 genomic region, and (**D**) the TAA6 genomic region, using either the GTR + G (A, B, D) or GTR + I (C) models. The size of each nucleotide alignment is given in parenthesis. Bootstrap values greater than 50% are indicated on nodes of each phylogenetictree. GenBank sequences are indicated in parenthesis, and sequences from this study are in bold font. The phylogeny (**A**) is rooted with *Pratylenchus penetrans* as an outgroup. Scale bar represents substitution per nucleotide position. For increasing the readability of panels (**B**–**D**), names of *M. graminicola* and *M. oryzae* are distinguished by green and blue, respectively.

**Figure 4 genes-10-00175-f004:**
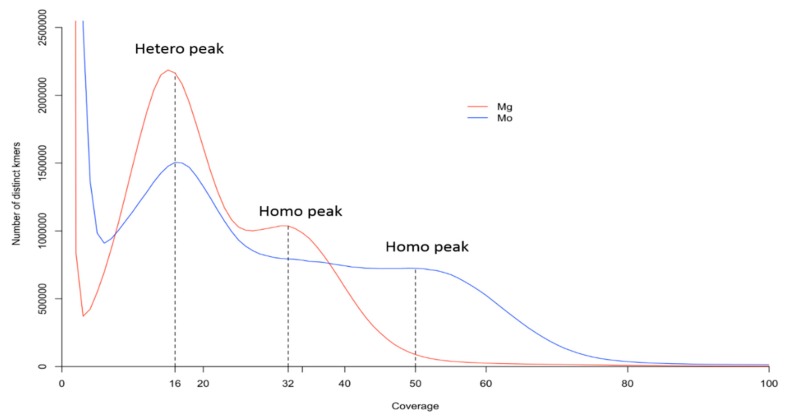
The 25-mer distributions of *M. graminicola* (red) and *M. oryzae* (blue) genomes. The dashed vertical lines indicate the possible positions of hetero- and homo-peaks.

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
