# Peer review of "On the Close Relatedness of Two Rice-Parasitic Root-Knot Nematode Species and the Recent Expansion of Meloidogyne graminicola in Southeast Asia"

_genes, 2019, doi:10.3390/genes10020175_

Round 1
Reviewer 1 Report
Besnard et al. have collected and sequenced isolates from 2 Meloidogyne rice-parasitic species M. graminicola (12 isolates) and M. oryzae (2 isolates). Then, they assembled the mitochondrion and 3 nuclear regions to assess the intra-species divergence of M. graminicola, the phylogenetic relationships of the two taxa inside the genus Meloidogyne, and the genomic structure of these two species. In addition, they performed K-mer analyses to explore ploidy levels.
The analyses of the mtDNA indicate a recent expansion of M. graminicola, while the phylogenetic inference validates the close relationship of these two species despite their different mode of reproduction. These two results along the produced sequencing data are an exciting start to dig deeper into their similarities and differences in an evolutionary context (sexual/asexual reproduction, meiotic/mitotic parthenogenesis).
Although the previous analyses were well performed, I have concerns about the interpretation of the results especially about the conclusion that both species are of hybrid origins. I believe that extra work needs to be done in order to verify this hypothesis. I think that the dataset (whole genome sequence of 14 isolates) is clearly under-analysed and that a better knowledge of the genomic structure could be reached.
I recommend this article for publication if most parts are re-written to tone down the hybrid hypothesis or additional analyses are conducted to provide more evidence.
Major comments:
- Are both species of hybrid origin?
Although that may be the case, the evidence presented here is not strong enough to validate it.
1) Considering the K-mer distributions, in the case of M. graminicola, this distribution can be observed due to a heterozygous genome as stated in the text. In the case of M. oryzae, increased ploidy levels will not produce a heavy tail in kmer-distributions. A triploid species will have a distribution with 3 distinct peaks. As to why this tail is observed, there can be multiple reasons a) data is not clean, b) bias in Illumina library, c) lots of repeated kmers at different levels etc. I would suggest repeating this analysis with Mo-M1 sequencing data to check if the distribution differs. On that note, it would be interesting to calculate the distribution for all isolates.
2) Considering the diverged genomic sequences, these can be present because of heterozygosity.
3) The nrDNA region contains only a few markers in regions that are more diverse in general. Is the distribution of markers across the whole region or are they located in a few hotspots? Is the sequencing depth equal for these two haplotypes?
4) Considering M. graminicola being the parent of M. oryzae, I think the data strongly suggest sharing a recent common ancestor, and if M. oryzae is a hybrid, then M. graminicola could be one of the parents. Based on the mitochondrial percentage identity it may be that M. graminicola was the male parent.
- Further analyses
1) The authors should be more careful when dealing with possible contamination. Although they performed an initial QC, I do not think that it is sufficient without any knowledge of the possible contaminants. Given the data, the authors can perform an initial assembly with SPAdes (or minia for speed) and do the QC with blobtools (https://github.com/DRL/blobtools).
2) Using these assemblies, the authors can also calculate the divergence between different copies of the genome rather than just a few regions. Furthermore, they can test the possibility of M. graminicola being the parent by checking the divergence between the two species.
3) Validating the hybrid origins is more difficult since there are no genome assemblies for closely related taxa (e.g. M. chitwoodii, M. naasi). One first step is to run CEGMA and BUSCO on the clean assemblies (step 1) and then check duplication levels.
Minor comments:
1) L. 42: Can the authors provide a citation for the number 80%?
2) L. 102: Indicate number of isolates per species
3) L. 136: Explain the annotation pipeline of M. graminicola and M. oryzae mtDNA
4) L. 147: What does it mean that the tree was validated with raxmlGUI?
5) L. 186-189: This sentence should go to 3.5
6) L. 199-202: The number of remaining reads after contamination removal should be added to Table S1
7) L. 205-206: The kmer plot with multiple values should also be done for M. oryzae
8) L. 267: Bootstrap support of 78 is not low
9) L. 307-309: Why didn't the authors use the whole length and only used the D2-D3 and ITS regions?
10) L. 337-338: According to Table S1 Mg-VN18 has 9.2 and Mo-M2 has 8.3 million pairs.
11) L. 346: Typically heterozygosity of >1% can produce a binomial distribution. I would prefer instead of "high heterozygosity" to be phrased like "more than 1% heterozygosity"
12) Table S1: It should be clarified that the sequence depth is for mtDNA; A new column should be added with genome sequence depth assuming the genome size estimated earlier
Grammar comments:
1) All species names in Introduction should be in italics.
2) L. 48: "In considering only Asia," -> "In Asia,"
3) L. 65: "hugediversity" -> "huge diversity"
4) L. 72: "TropicalAmerica" -> "tropical America"
5) L. 100: missing header "2. Materials and Methods"
6) L. 197: "Trimmomatics" -> "Trimmomatic"
7) L. 264: "lineages usually recognized" -> "lineages"
8) L. 301: "LS9744433" -> "LS974433"
9) L. 359: "were finally assembled" -> "were assembled"
10) L. 415: "M. florensis" -> "M. floridensis"
11) Table S1: There is "`" in the last column of second row
Author Response
We thank the referees for the constructive comments. As suggested by referee 1, we decided to tone down the hybrid hypothesis in the present version of the manuscript. All our responses to comments are attached in the pdf file.

Reviewer 2 Report
line 21 “how this parasite successfully colonized so many different environments ” , modified to “how this nematode successfully damaged rice in many different environments ” modified to
line 46 economic loss of over euro 70 billions is wrong , check the reference 2
line 97 line 98 the nematode name should be in italic
line 214 the nematode name should be in italic
Author Response
responses to referee comments in the attached file.

Round 2
Reviewer 1 Report
The authors have addressed all my comments